# What are the perceptions and concerns of people living with diabetes and National Health Service staff around the potential implementation of AI-assisted screening for diabetic eye disease? Development and validation of a survey for use in a secondary care screening setting

Kathryn Willis  ,[1] Umar A R Chaudhry,[1] Lakshmi Chandrasekaran,[1] Charlotte Wahlich,[1] Abraham Olvera-Barrios,[2] Ryan Chambers,[3] Louis Bolter,[3] John Anderson  ,[3] S A Barman,[4] Jiri Fajtl,[4] Roshan Welikala,[4] Catherine Egan,[2] Adnan Tufail,[2] Christopher G Owen,[1] Alicja Rudnicka,[1] On behalf of the ARIAS Research Group

For numbered affiliations see end of article.

**Correspondence to**
Dr Alicja Rudnicka;
arudnick@sgul.ac.uk

## ABSTRACT

**Introduction** The English National Health Service (NHS) Diabetic Eye Screening Programme (DESP) performs around 2.3 million eye screening appointments annually, generating approximately 13 million retinal images that are graded by humans for the presence or severity of diabetic retinopathy. Previous research has shown that automated retinal image analysis systems, including artificial intelligence (AI), can identify images with no disease from those with diabetic retinopathy as safely and effectively as human graders, and could significantly reduce the workload for human graders. Some algorithms can also determine the level of severity of the retinopathy with similar performance to humans. There is a need to examine perceptions and concerns surrounding AI-assisted eye-screening among people living with diabetes and NHS staff, if AI was to be introduced into the DESP, to identify factors that may influence acceptance of this technology.
**Methods and analysis** People living with diabetes and staff from the North East London (NEL) NHS DESP were invited to participate in two respective focus groups to codesign two online surveys exploring their perceptions and concerns around the potential introduction of AI-assisted screening. Focus group participants were representative of the local population in terms of ages and ethnicity. Participants' feedback was taken into consideration to update surveys which were circulated for further feedback. Surveys will be piloted at the NEL DESP and followed by semistructured interviews to assess accessibility, usability and to validate the surveys.
Validated surveys will be distributed by other NHS DESP sites, and also via patient groups on social media, relevant charities and the British Association of Retinal Screeners. Post-survey evaluative interviews will be undertaken among those who consent to participate in further research.
**Ethics and dissemination** Ethical approval has been obtained by the NHS Research Ethics Committee (IRAS ID: 316631). Survey results will be shared and discussed

## STRENGTHS AND LIMITATIONS OF THIS STUDY

⇒ Survey content has been codesigned with focus group participants representing people living with diabetes and healthcare professionals which contributes towards content validity of the survey as all questions have been reviewed by relevant populations.

⇒ Face validity of survey content has been provided through expert review of the survey by clinicians and collaborators from the North East London Diabetic Eye Screening Programme.

⇒ At present, the survey for people living with diabetes is only available in English and has not currently been translated into any other language which could be a limitation given the diverse population of individuals who will be invited to take part, however, we have attempted to mitigate this by recommending the help of a friend or family member who is able to translate.

⇒ As the survey requires people to be able to read and understand English, there would also be limitations surrounding literacy level, whereby a friend or family member may be required to read the survey to the participant.

⇒ The online platform of the survey could limit participation by people living with diabetes if they do not have access to the internet or a device to complete the survey.

with focus groups to facilitate preparation of findings for publication and to inform codesign of outreach activities to address concerns and perceptions identified.

## INTRODUCTION

There has been an increasing body of literature in the UK to encourage the safe use of artificial intelligence (AI) in healthcare as a means of informing decision-making and combatting a workforce shortage by automating tasks, driven by governmental bodies and large organisations such as Health Education England and the National Health Service (NHS).[1–4] This could provide huge benefits for primary, secondary and tertiary care services and healthcare professionals (HCPs), by allowing staff to spend more time with patients or on diagnosis of more severe or subtle disease, by removing repetitive or low risk tasks.

One area of secondary care which could see immediate benefits from the implementation of AI is diabetic eye screening. The English NHS Diabetic Eye Screening Programme (DESP) performs over 2 million eye screening appointments each year, generating approximately 13 million retinal images.[5] These images are assessed by up to three trained human graders for the presence and severity of diabetic retinopathy (DR), and those with potentially sight-threatening diabetic eye disease are referred to Hospital Eye Services. As the number of poeple living with diabetes is increasing, this represents a major challenge to healthcare providers. Emerging automated retinal image analysis systems (ARIAS) including AI using machine learning algorithms may provide cost-effective alternatives to a purely human grading system.[6]

The research team have previously shown that ARIAS could be used to triage those at medium high risk of sight-threatening DR, from those at low risk of sight-threatening retinopathy, thereby reducing the need for all screening episodes to receive human grading and providing significant cost savings for the NHS.[7–9] In April 2021, an external review commissioned by the UK National Screening Committee and Public Health England recognised our previous work in this area as being of high methodological quality with direct applicability to the NHS DESP.[6]

Automated image grading by computer algorimths (inluding AI) as a 'first pass' is currently being used in some countries, including Scotland and Portugal, and is being considered by other countries including the USA, the Netherlands and Spain.[6] Although no ARIAS are currently approved for use in the English NHS DESP, a recent evidence synthesis review by Zhelev et al[6] recommended a staged implementation of a commercially available ARIAS previously evaluated by our study team.[7–9]

There is a general consensus that medical policy around AI in healthcare should be driven by patient and public health outcomes, and that evaluations of test performance alone are insufficient.[10 11] Before AI-assisted screening can be implemented into the English NHS DESP, it is essential to understand how this technology would be received by NHS staff/HCPs working in the DESP, and people living with diabetes that attend eye screening. The study outlined in this paper is specifically aimed at examining views of HCPs and people living with diabetes regarding the potential use of AI-assisted screening in the English NHS DESP.

There is a breadth of research on the general acceptability of clinical AI from the perspective of healthcare workers,[12] and patients' perspectives on clinical AI within the fields of oncology,[13] radiology,[14 15] dermatology[13–15] and ophthalmology[16–18] that could revolutionise certain aspects of patient care, such as screening for DR.

### Clinical AI and the HCPs' perspective

At the time of writing, there has been no research carried out in England specifically among English NHS DESP staff regarding their perceptions towards potential implementation of AI within the DESP.

A qualitative exploration through individual interviews was carried out by the NHS AI Lab to identify factors that affect the confidence of HCPs in using AI technologies.[1] This research identified trust, governance and the importance of sufficient training for HCPs as key themes to build confidence in incorporating AI technology into practice.[1] The current study will build on these themes and quantify opinions in relation to the English NHS DESP.

Elsewhere, a study based in Malaysia exploring perceptions and concerns of clinicians about the use of AI in medicine found that, while 78.3% (88/112) of participating staff agreed that AI can improve the speed of healthcare processes, 81.7% would prefer to follow the opinion of a doctor rather than that of AI,[19] implying low confidence in AI among clinicians in this study from a variety of fields. However, 29.5% were working within a surgical-based setting including radiology, pathology, dermatology and ophthalmology and their views may not be representative of other clinical subspecialties.

Scheetz et al[13] conducted a survey into the thoughts and opinions from clinicians about AI from ophthalmology, dermatology, radiology and radiation oncology in Australia and New Zealand. Their findings agreed with others[19 20] in terms of improving the speed of monotonous healthcare processes. Other benefits included improved patient access to screening and improved diagnostic confidence. The study found ophthalmologists were more than twice as likely to use AI in their daily clinical practice compared with radiologists and dermatologists (15.7% compared with 6.1% and 5.2%, respectively), and only 41.6% of ophthalmologists felt AI would impact the workforce to 'a great extent'. This could be due to the current familiarity, and in some cases implementation, of this technology. However, the need for further training and education around clinical AI if it was to be implemented has been acknowledged.[13 21 22]

There is a need to assess and address concerns around AI-assisted screening among clinical and non-clinical staff working in the NHS in England. The present study aims to explore perceptions of NHS staff (in addition to

people living with diabetes) and identify potential barriers or enablers to the acceptance of AI-assisted screening in the English NHS DESP among HCPs working in the screening programme.

## Clinical AI and the service user perspective

Regarding implementation of AI into the English DESP, there has been no prior research carried out to assess the attitudes or how AI-assisted screening would be received among people living with diabetes in England. Public confidence in AI technologies being used in healthcare was highlighted by NHS staff as being necessary for successful implementation in healthcare settings, and the need for public engagement and education outreach was emphasised.[1]

A recent systematic review identified 23 research studies that assessed the attitudes of patients and the general public towards clinical AI.[23] Six key themes were identified across all research studies included in the review: AI concepts; acceptability; relationship with humans; development and implementation; risks; strengths, benefits and weaknesses. These themes also resonate with research conducted into the attitudes of HCPs discussed above.

Previous research into perceptions of AI-assisted screening for diabetic eye disease from other countries is sparse, but limited evidence has implied high acceptability among people living with diabetes,[16 18] with one Australian study reporting up to 96% (92/96) of participants were either satisfied or very satisfied with the automated screening model. At 1-month follow-up, 78% (43/55 of contactable participants) stated that they preferred AI-based eye screening to manual screening, however trust in AI was listed as a main reason for preference for a manual screening model. This research focused mostly on the impact of AI on the screening process, as opposed to perceptions and concerns of participants about the use of AI technology.[16] A study in New Zealand[18] found that 78% of 438 study participants were comfortable with the potential use of AI in diabetic eye screening, despite only 58% having heard of AI being used in healthcare, and only 59% of participants were aware that AI could be used in a clinical setting to assist with diagnosis. Survey responses were found to differ slightly based on age and/or ethnic group,[18] which further supports the importance of the current study to understand opinions across a different population of service users in the English NHS DESP.

Since the studies mentioned above took place in Australia or New Zealand, the findings may not be directly applicable to the English population where there are differences in population demographics. There is a need to identify barriers and enablers among NHS DESP staff and people living with diabetes to the acceptance of AI-assisted screening within the English NHS DESP, so that appropriate resources and outreach activities can be considered that would facilitate implementation, provide reassurance and retain engagement with the service at a similar level. These barriers and enablers are not only relevant to the current study, but have been identified as being of importance at a UK government level.[6]

## Rationale

It is clear from the literature that there is a need to investigate how the implementation of ARIAS/AI-assisted screening into the DESP would be received by HCPs and people living with diabetes. The overarching aim of this study is to codesign surveys that examine the existing knowledge base, perceptions and concerns among NHS staff and people living with diabetes (both type 1 and type 2) from a diverse sociodemographic and geographical location prior to the introduction of AI-assisted screening for diabetic eye disease into the English NHS DESP. As people from different age and ethnicity groups may have differing opinions towards AI in diabetic eye screening, it is vital to ensure representative samples from different age and ethnic groups are purposely sought. Examining differences in views and acceptance of this technology between different subgroups of the population is a focus of this study. It will assist with the translation of this technology into real-life screening programmes and minimise the potential for inequitable care across different subgroups of the population as a result of ARIAS deployment, irrespective of its accuracy or test performance. At the time of writing, no such survey has been undertaken within the NHS DESP to address these issues.

## AIMS AND OBJECTIVES

This research aims to collect qualitative and quantitative data from HCPs that work in the English NHS DESP and people living with diabetes that interface with the English NHS DESP about their views on the future incorporation of AI-assisted screening.

Specific aims are to:

1. Investigate patterns in perceptions and concerns among NHS staff and adults living with diabetes if AI-assisted screening were to be introduced into the NHS DESP in future.
2. Examine differences in patterns by age, sex, ethnic group and sociodemographic indicators such as Index of Multiple Deprivation (IMD) and by English NHS DESP location.
3. Investigate differences in patterns between NHS staff and people living with diabetes.
4. Identify barriers and enablers to the acceptance of this technology within the English NHS DESP among staff and people living with diabetes.
5. Synthesise the evidence from (1 to 4) to generate evidence-based recommendations to optimise the transition phase if AI were to be introduced into the English NHS DESP.

## METHODS AND ANALYSIS
### Patient and public involvement
The present research is a patient and public involvement project for a wider body of research investigating how ARIAS could be used in diabetic eye screening for DR. People living with diabetes and NHS staff working in the DESP have been actively involved in the present research to date. Establishing and accurately representing their views and priorities was a key part of the codesign of our survey and in formulating this protocol.

### Survey codesign
The term codesign has been adopted to explain our methodology as an umbrella term which covers codevelopment. The methodology has been designed in accordance with the UK Standards for Public Involvement and the Ethnicity framework. These frameworks ensured that representatives were suitably involved in the survey design process so that survey questions were inclusive and appropriate for each cohort, which ensured the overarching patient and public involvement aims for this research were met. The Checklist for Reporting Results of Internet E-Surveys[24] was also used to assist survey design due to its focus on how to report important findings from web-based surveys.

The research team reviewed recently published literature aimed at exploring staff and public perceptions of AI in a screening setting. We identified seven surveys which explored the views of HCPs,[12 13 19 25–28] and five surveys which explored the views of the public and service users.[16 18 29–31] These survey questions were collated by the research team and considered against the objectives to identify any relevant survey questions which could be beneficial for this research. Novel survey questions were also created in line with study aims to develop preliminary questions framed in relation to diabetic eye screening. These draft questions provided a structure for focus group discussions.

Focus groups participants were recruited between October and December 2021 by the clinical lead at the North East London (NEL) DESP (JA) using purposive sampling to represent a diverse mix of people living with diabetes and HCPs, respectively. The project team approached individuals via email, provided a brief summary of the project, study design, what participation in focus groups would entail (particularly emphasising involvement in codesign of questionnaires for wider dissemination), and the opportunity for participants to ask any questions before the first focus group was scheduled. Participants in the HCPs focus group were four men and three women, and participants from the people living with diabetes focus group were two men and four women. Both focus groups were balanced in terms of age (spanning a wide age range) differing roles within the DESP as well as capturing the ethnic diversity of NHS DESP service users and staff.

Focus group sessions took place online via Microsoft Teams between March and June 2022 to encourage ease of access for participants. As per the UK standards for public involvement, focus group sessions followed an agenda whereby the purpose of involvement in focus group sessions was established and defined, and the potential benefits of taking part were highlighted including payment, discussion and interpretation of survey findings and active participation in publications emanating from the work.

Both the survey lay summary and draft survey questions were openly discussed and refined in focus groups with the research team. Questions were then updated and shared with focus group participants by email to ensure all feedback had been addressed, and additional comments from focus group participants were incorporated to ensure surveys accurately reflected the views of each cohort. This iterative process continued until HCPs and people living with diabetes had no further feedback on the content or structure of the surveys.

### Results from focus group sessions
The two final approved surveys were comparable in their content and took an average of 10 minutes to complete. The survey for HCPs comprised 21 questions, and the survey for people living with diabetes comprised 28 questions. The questionnaire features both negatively and positively worded items within the same group of questions, which were designed to account for acquiescence bias. Both surveys include sociodemographic questions (eg, age, sex, ethnicity, level of education), general questions about using AI for eye screening and focused questions that can be categorised into key themes which were identified and finalised by the focus groups (table 1). The updated surveys were redistributed to focus groups to ensure participants were happy with the layout and to ensure participant feedback had been accurately represented in the final survey version. The majority of questions about AI-assisted eye screening used a 5-point Likert

**Table 1** Key themes examined in surveys for HCPs and people living with diabetes

| HCPs | People living with diabetes |
| --- | --- |
| ► About you | ► About you |
| ► Technology in your daily life | ► Technology in your daily life |
| ► General Qs about AI for eye screening | ► General Qs about AI for eye screening |
| ► Efficiency | ► Efficiency |
| ► Data regulation/security | ► Data responsibility/security |
| ► Trust | ► Trust |
| ► Impact on workforce | ► Screening experience |
| ► Screening experience and patient–practitioner relationship | ► Happy for us to 'stay in touch' with follow-up survey? |
| ► Happy for us to 'stay in touch' with follow-up survey? | |

AI, artificial intelligence; HCPs, healthcare professionals.

scale from 'strongly agree' to 'strongly disagree'. Focus group participants were familiar with the use of Likert-style questions in other settings and this question style has been considered as one of the best measurement method in psychological research.[32]

## Focus group experiences

We received a range of feedback from HCPs and people living with diabetes about their participation in our focus groups. Our primary aim for our focus groups was to be open and respectful of participants thoughts and opinions as we discussed the subject and the survey design. All of our participants found the focus group sessions interesting and engaging and enjoyed being a part of the survey codesign process. Please see box 1 for anonymous feedback from participants—there are some useful points that we will consider in further qualitative validation work and in phase 4 when we conduct post survey interviews.

## Online survey platform

To distribute the survey widely and gather data in an efficient and accurate manner, Jisc Online Surveys platform (onlinesurveys.ac.uk) will be used to host the survey. This was selected due to its ease of use on computerised or mobile devices, its ability to securely store and export survey responses, mark questions as mandatory to prevent data loss and allow key study documents such as the participant information sheet (PIS) to be incorporated. The usability and technical functionality of the survey platform on different devices including mobile phones and laptops was initially tested by the research team and then distributed to respective focus groups to trial on different devices to confirm that this was a usable platform.

The first page of the survey outlines what the research is about, why the survey is being done and what we hope to achieve. The first page also includes a statement about consent to participate in an anonymous survey, what participation in the survey will involve and a separate link to the PIS. While the surveys are open, participants will be able to complete the survey at any point during this time frame. Participants will be expected to complete the survey only once.

## Eligibility criteria

For the patient survey, participants must have been diagnosed with diabetes and have recently completed, at least, one diabetic eye screening appointment in the English NHS DESP within the past 3–4 months, at the time of survey distribution. For both the patient and NHS DESP staff survey, participants should have access to a computer or mobile phone with internet access to complete the survey. The surveys were written in English, and participants must be willing to participate and understand that consent is implied by completion of the survey.

Participants <16 years of age, and participants without access to a device with Internet would be excluded from participating. Inability to understand informed consent or people with cognitive impairment will not be eligible.

---

**Box 1** Anonymous feedback from people living with diabetes and healthcare professional who participated in focus groups.

**Feedback from people living with diabetes**
'I felt the focus group was well organised and informative. In hindsight, it would have been good to include a patient from the >65 age group given this demographic may find the introduction of AI screening more abstract to identify with. Although it was helpful to learn AI screening is already in use in Scotland and other European countries. The sessions went quite well and were interactive in discussions and I felt the leads were really taking on board the feedback provided by the focus group. This was also reflected when the subsequent drafts incorporated these updated comments. I also liked the fact that for every question presented the rationale was provided and as we progressed through the sessions the layout and order was changed and I feel this made for a good final survey.' **Participant living with diabetes, participant 1**

'I've been involved in several focus groups in the past but this was one of the best. I felt like my views were really important, as if I was part of the team.' **Participant living with diabetes, participant 2**

'I was really impressed with how the team spoke to the participants in the focus group. We were really encouraged to say what we thought. Our responses were captured and then translated into the next iteration of the survey. I lost count how many drafts we had because the team were keen to make sure the final survey really reflected everything we felt needed to be there. They were tenacious!' **Participant living with diabetes, participant 2**

**Feedback from HCPs**
'I felt the survey and focus groups were accessible. Being able to join the focus groups over teams from our individual locations was essential to me partaking in the study. There was also adequate time from receiving the survey to needing to provide feedback, again improving accessibility and allowing proper evaluation of the survey to take place.' **HCP, participant 1**

'The focus group session especially made it feel like a co-designed survey as the team was easy to talk to and engaged with all ideas that were provided by facilitating open ended discussions. We would see changes implemented in the next iteration based on these discussions such as altering the structure of the survey, the balancing of questions and including a request for future research. This highlighted to me that our input was valued and that the survey was being developed by a partnership between the study team and the focus group.' **HCP, participant 1**

'AI in general is a very interesting subject so to be involved in a focus group related to AI and DESP was a pleasure. It will be intriguing to see what impact AI has on screening, grading and DESP service as a whole going forward.' **HCP, participant 2**

'I came to the Focus Groups with a fairly established personal view on how I saw the role of AI in screening. What was interesting about the Focus Group was discussing my opinions in the bigger context of other views that did help me understand the range of views from completely relying on AI to not using it at all. As a result, I could see the value of designing a questionnaire that would be able to elicit the range of views so that an approach to implementing AI assisted grading could take into account the positives and negatives that the survey would produce.

Continued

## Box 1 Continued

Overall I found the process to be engaging, deepening my own understanding of the issues and broadening the context in which embedding this into DESP entails.' **HCP, participant 3**

'Taking part in this focus group was both interesting and enlightening in that some of the topics that came [up] were ones that I hadn't really thought about in relation to AI until we discussed them as a group. In particular how different ethnic groups may view AI (both staff and patients) and how important it is that the AI is reliable in recognising diabetic retinopathy in different ethnic groups, which will make it more appealing to all. I particularly like that there is a phase III of distributing the survey to hard-to-reach groups, who are an often forgotten group, but one which many of our patients will fall into.' **HCP, participant 4**

'It was interesting to see the different viewpoints people held towards AI but it seemed the overarching opinions towards it were more cynical. I found it initially hard to understand the use of AI in healthcare and felt that it would not be beneficial, however after discussing it further I started to see the benefits of it. It seemed there was a concern for impact on workforce and how it may affect jobs. But I also did start to see that using AI would allow graders to focus on the more severe or complex cases of diabetic retinopathy.' **HCP, participant 5**

### Survey distribution

The survey URL will be distributed via participating NHS Diabetic Eye Screening Centres in England and other routes using a three-phase approach (figure 1). This will be followed by a fourth phase of postsurvey evaluation.

### Phase I

During phase I (figure 1A), surveys will be launched at the NEL, which will act as a pilot to confirm feasibility and functionality of the online platform and recruitment method.

As shown in figure 1, NHS staff and people living with diabetes from the NEL DESP will be recruited by the clinical lead or programme manager. NHS staff will be recruited by an email, while people living with diabetes will be recruited by a text messaging service sent by the programme manager which is an established form of communication with patients as it is used to send appointment reminders and service feedback surveys. Additionally, a poster has been created with a QR code to the survey link that could be placed in the waiting area where people living with diabetes attend for screening.

Participants who complete the survey and volunteer to provide their contact details for participation in further research will be contacted to participate in semistructured qualitative interviews to validate the surveys. This validation step will include discussions around comprehension of survey questions, readability, ease of use of the online platform using different devices, relevance of questions and whether additional questions are needed. The surveys will be reviewed and refined where necessary. After validation, survey dissemination at the NEL DESP will continue for 3 more months to all HCPs and people living with diabetes that recently attended for screening.

### Phase II

Phase II (figure 1B) will involve wider distribution of the surveys through other invited NHS DESP centres. Purposive sampling has been used to recruit centres which are geographically dispersed. Recruitment of participants will take place either via text messaging service sent by the programme manager, or by a handout with the survey URL and QR code which will be distributed by NHS staff during appointments.

### Phase III

Phase III (figure 1C) seeks to further boost response rates from people living with diabetes that are from hard-to-reach groups by distributing the survey through multiple charities. Patient representatives from each NHS DESP will also be approached to help disseminate the survey link to local patient groups and related social media. The British Association of Retinal Screeners has agreed to distribute the survey to a wider group of HCPs via established online communications.

### Phase IV

Participants who provide their contact details for participation in further research will be invited to participate in short qualitative interviews to evaluate the effectiveness of the survey in meeting the original aims and objectives and to discuss next steps for outreach activities based on the results (figure 1D). It is important to evaluate the survey among participants that have completed the survey so that if the research was to be repeated elsewhere or on a larger scale, any additional feedback could be considered and incorporated into future repeat surveys. We aim to contact three people living with diabetes and two HCPs from each recruitment pathway.

### Sampling and representativeness
#### People living with diabetes cohort

The target sample size for people living with diabetes is 300 participants. The NEL DESP will be our primary recruitment site for people living with diabetes. It is one of the largest screening centres encompassing a wide spectrum of people living with diabetes (~120 000 people invited annually), and is one of the most ethnically diverse screening centres in England (white 29%, black 17% and South Asian 41% ethnicity).[33] The NEL DESP screens approximately 10 000 participants each month and all eligible adults would be contacted after their recent screening appointment. If response rates are low, for example, 1%, we should achieve the desired sample size for people living with diabetes over a 4-month period.

When the validation work as a part of phase I is complete, we will simultaneously start recruitment via other geographically dispersed DESP centres, charities and local patient groups which we anticipate will increase the total number of responses, including those from ethnic minority groups. This will enable the stability and generalisability of survey findings across different population groups across England to be summarised. Where

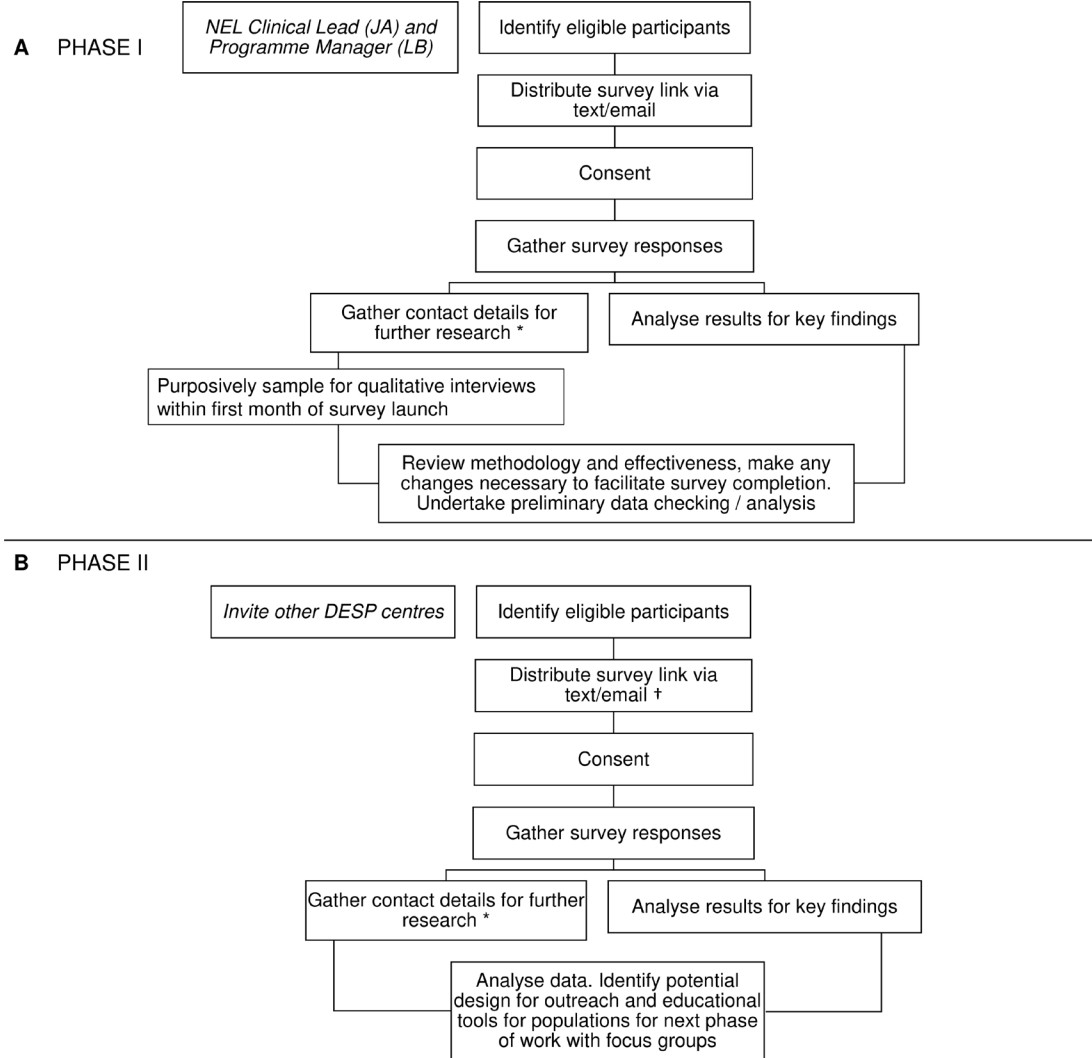

**Figure 1** Survey distribution flow diagram and postsurvey evaluation. *Indicates that this stage is optional—participants can volunteer to provide contact details but it is not mandatory. †Indicates that other recruitment methods may be used dependant on the DESP site—any recruitment materials will be submitted as an amendment for ethical approval to the Health Research Authority (HRA). DESP, Diabetic Eye Screening Programme; NEL, North East London; NHS, National Health Service.

possible, response rates will be monitored using unique survey URLs for each recruitment pathway together with the corresponding number of invites.

We are optimistic that recruitment of at least 100 people living with diabetes from each of the three main ethnic groups should be achievable within a reasonable time frame of 4 months.

### HCP cohort

The number of HCPs per NHS DESP centre varies from approximately 30 to 90 per centre. To achieve the desired sample size of 300 for HCPs, we aim to recruit from a minimum of five NHS DESP sites and via the British Association of Retinal Screeners where the majority of HCPs working in the NHS DESP are registered.

### Sample size calculation for people living with diabetes and HCP cohorts

Analyses will be undertaken within each survey domain, including (1) general questions about AI for screening, (2) efficiency, (3) data regulation and security, (4) trust, (5) screening experience and an additional domain covering (6) impact on the workforce for NHS staff survey. One hundred participants per ethnic subgroup will allow approximately a one-step difference in median Likert scores to be detected between groups (assuming a common SD of 1.5) with 95% power and alpha (type 1 error) set to 0.01. Data collected on sociodemographic factors (eg, IMD) will be grouped into a maximum of 5 subgroups for analytical purposes, providing

approximately 60 participants per subgroup. This reduces the power to between 90% and 95% and alpha to 0.05 to detect one-step difference in median Likert scores between subgroups.

## Analysis plan

Data from the surveys will be automatically collected and securely stored by the survey platform and exported for analysis (eg, Microsoft Excel csv). Survey responses will be presented visually by domain using a combination of radar plots and bar charts, stratified by age quartiles, ethnic group, sex, geographical location (by DESP centre) and IMD quintiles. Data will be summarised using descriptive statistics using percentages or medians.

For the responses to the 5-point scale measuring agreement or disagreement across a series of questions in each domain, average scores will be presented as a measure of agreement with a series of belief statements (means or medians depending on distribution of the data). Spearman correlation will be used to examine the association of Likert scores with age, level of education and IMD. Differences in Likert scores between population subgroups of age, sex, ethnicity, level of education, geographical location, IMD and persons with type 1 vs type 2 diabetes will be formally compared using Mann-Whitney U test. If sufficient responses are received and the data are normally distributed, parametric tests will be used. Likert choices 'strongly agree' and 'agree' will be combined into an overall per cent agreement score (ie, the percentage of the sample that agreed with the statement). $\chi^2$ tests will be used to quantify statistical differences in percent agreement responses between population subgroups of age, sex, ethnicity, level of education, geographical location, IMD and persons with type 1 vs type 2 diabetes. Survey responses for similar questions will be compared between HCPs and people living with diabetes.

A key focus of the analysis throughout will be to identify enablers and barriers to the acceptance of AI-assisted screening for diabetic eye disease. Exploratory factor analysis will be used to examine if questions can be mapped to distinct scales that identify a set of latent constructs underlying the questionnaire responses. Internal reliability of the scales will be assessed using Cronbach's alpha. All analyses will be undertaken using STATA V.17 (Stata).

## ETHICS

This research has been ethically approved by the NHS North West - Haydock Research Ethics Committee and the Human Research Authority (reference: 22/NW/0402; IRAS project ID: 316631).

## DISSEMINATION

Findings from the surveys will be shared and discussed with our focus groups. We will continue to recruit more participants for focus groups by contacting those who provide contact details after completing the survey. Together we will summarise the survey findings and explore wider implications. Findings will be shared with all partners involved in survey dissemination prior to submission for publication (when the survey questions used will be shared).

The next stage would involve codevelopment of communications and dissemination materials in collaboration with our focus groups to establish the best methods to share our results with the wider screening community (including NHS quality assurance and commissioning teams) and with policy-makers.

Suggestions for communications about AI in the DESP from HCPs who took part in reviewing this paper included online training, diabetic education classes for staff and people living with diabetes, and traditional outreach approaches in the form of leaflets, posters, presentations and videos. Suggestions from people living with diabetes included infographics and videos to be shared via social media, radio and sharing information with diabetes charities to add to their resources for people to access.

Survey findings will also inform the codesign of information outreach for people living with diabetes and NHS staff to assist the transition from human-led screening to AI-assisted screening.

## DISCUSSION

This section covers the strengths and weaknesses of the current study design.

### Strengths

One key strength of this research is the collaborative codesign of the surveys with our target population groups, that is, HCPs and people living with diabetes. This has ensured content validity, as each of the questions and survey versions have been reviewed and approved by our focus groups. A further validation stage to confirm usability and functionality of the surveys provides additional strength to the study design. Both surveys have also been reviewed by the research team and by our steering advisory group which includes clinical and non-clinical academic staff. We have deliberately targeted two large urban NHS screening centres with diverse population groups to capture the views from a broad spectrum of individuals from different ethnic and other sociodemographic population subgroups.

### Limitations

Due to the methods used to recruit people living with diabetes and NHS staff, we are aware that there will be limitations in terms of potential selection bias and representativeness of the survey population. This includes potential under-representation of people where English is not their first language, unless support can be offered by family members and friends to assist with survey completion. There may also be under-representation from people who do not have access to the internet or an

electronic device to complete the survey, which may have a greater impact on older adults. Using electronic distribution and completion of surveys may impact on survey response rates compared with using electronic and paper survey formats for distribution and completion.

**Author affiliations**
[1]Population Health Research Institute, St George's University of London, London, UK
[2]NIHR Biomedical Research Centre, Moorfields Eye Hospital NHS Foundation Trust, London, UK
[3]Diabetes and Endocrinolgy, Homerton Healthcare NHS Foundation Trust, London, UK
[4]School of Computer Science and Mathematics, Kingston University London, London, UK

**Correction notice** This article has been corrected since it was published. Study Advisory Group members have been removed from the author list.

**Collaborators** The Artificial Intelligence / Automated Retinal Image Analysis Systems (ARIAS) Research Group: John Anderson, Sarah Barman, Louis Bolter, Ryan Chambers, Lakshmi Chandrasekaran, Umar Chaudhry, Karen Easy, Cathy Egan, Jiri Fajtl, Tasmina Hussain, Rizwana Issa, Aaron Lee, Fiona Martin, Peter Mitchell, Abdul Mulla, Gbenga Olasehinde, Abraham Olvera-Barrios, Christopher G Owen, Ahmed Patel, Oscar Phillips, Paolo Remagnino, Alicja R Rudnicka, Adnan Tufail, Charlotte Wahlich, Roshan Welikala, Kathryn Willis. Study Advisory Group members: Rosalind Given-Wilson, Alastair Denniston, Kevin Dunbar, Samantha Mann, Fiona Martin, Giuseppe Sollazzo.

**Contributors** All authors, including members of the *A*rtificial Intelligence and Automated *R*etinal *I*mage Analysis *Sy*tems - ARIAS research group consortium, contributed to this manuscript. KW, LC, UARC, CW, LB, RC, JA, CGO and AR drafted the study design. All coauthors contributed to iterative updates. KW and AR wrote the first draft of the report, which LC, UARC, CW, LB, RC, JA, AO-B, SAB, JF, RW, CE and AT contributed to and critically appraised. AR and CGO are responsible for data integrity and will act as guarantors.

**Funding** This work was supported by the Wellcome Trust (2022, 224390/Z/21/Z); St Georges, University of London Participatory Fund; NHS Transformation Directorate and The Health Foundation managed by the National Institute for Health and Social Care Research (AI_HI200008).

**Disclaimer** The views expressed in this publication are those of the author(s) and not necessarily those of the NHS Transformation Directorate, The Health Foundation, National Institute for Health Research, or the Department of Health and Social Care.

**Competing interests** None declared.

**Patient and public involvement** Patients and/or the public were involved in the design, or conduct, or reporting, or dissemination plans of this research. Refer to the Methods section for further details.

**Patient consent for publication** Not applicable.

**Provenance and peer review** Not commissioned; externally peer reviewed.

**ORCID iDs**
Kathryn Willis http://orcid.org/0009-0006-2847-3573
John Anderson http://orcid.org/0000-0002-2355-9742

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
