## [Reviewer comments · BMJ Open]

ARTICLE DETAILS

TITLE (PROVISIONAL)	What are the perceptions and concerns of people living with diabetes and NHS staff around the potential implementation of AI-assisted screening for Diabetic Eye Disease? Development and validation a survey for use in a secondary care screening setting
AUTHORS	Willis, Kathryn; Chaudhry, Umar; Chandrasekaran, Lakshmi; Wahlich, Charlotte; Olvera-Barrios, Abraham; Chambers, Ryan; Bolter, Louis; Anderson, John; Barman, S. A.; Fajtl, Jiri; Welikala, Roshan; Egan, Catherine; Tufail, Adnan; Owen, Christopher; Rudnicka, Alicja

VERSION 1 – REVIEW

REVIEWER	Ongena, Yfke P University of Groningen
REVIEW RETURNED	03-Jun-2023

GENERAL COMMENTS	The research described in the aims to collect qualitative and quantitative data from HCPs and people living with diabetes about their views on the future incorporation of AI into the NHS DESP. The rationale for the study is explained well and extensively. In the methods section, the development process of the questionnaire is generally clear, but it would be helpful to have a bit more details on the specific items used. Also, the authors write that majority of questions utilised a 5-point Likert scale from "Strongly agree" to "Strongly disagree". Such Likert-type agree-disagree (AD) scales suffer from response order effects, in the decremental AD scales (from positive agree to negative disagree), primacy effects are likely to occur (Höhne & Krebs, 2018). Especially if the items are worded positively, an acquiescence bias in responses is likely to occur. Therefore it is recommended to use an incremental scale or to use item-specific response options. The use of a focus group for development of a questionnaire in its first stage is obvious, but it is not clear to me why the focus group was used for final feedback on the questionnaire ("The surveys were redistributed to focus groups to ensure participants were happy with the layout and any suggested feedback was considered.") How did this redistribution take place? What do the authors mean with "feedback was considered"? Since mobile phones were also allowed as a device for survey completion: was this also taken into account when testing usability and technical functionality of the survey platform? On page 10, line 16 the authors write "expected to complete the survey only once"; it is unclear what this entails: was it technically possible to actually complete the survey more than once? Recruitment of participants in phase I is not entirely clear to me. Posters will be used, but how to make sure participants will have
--

	attention for these posters? This might introduce selection effects, depending on one's attentiveness to posters and experience in using QR-codes (or reluctance to do so). In the flow chart of figure 1 it is also mentioned that surveys will be distributed via text or email, but this is not further described in the main text. In phase IV, it is not clear to me what is meant by "effectiveness of the survey". Rather than extending the data collection period to 4 months (page 11, line 49/50) it could also be considered to increase response rates via mixed-mode surveys, for instance, paper-and-pencil versions for respondents reluctant to use electronic surveys. In addition, use of incentives (Piper et al., 2018) or personalisation (VanGeest & Johnson, 2012) may be considered. References: Höhne, J. K., & Krebs, D. (2018). Scale direction effects in agree/disagree and item-specific questions: A comparison of question formats. International Journal of Social Research Methodology, 21(1), 91-103. Pieper, D., Kotte, N., & Ober, P. (2018). The effect of a voucher incentive on a survey response rate in the clinical setting: a quasi-randomized controlled trial. BMC medical research methodology , 18(1), 1-4. VanGeest, J. B., & Johnson, T. P. (2012). Using incentives in surveys of cancer patients: do "best practices" apply?. Cancer Causes & Control , 23, 2047-2052.
--	---

REVIEWER	Vaghefi, Ehsan The University of Auckland, optometry and vision sciences
REVIEW RETURNED	12-Jun-2023

GENERAL COMMENTS	This is a clear description of the survey that is going to be performed in the community
--

VERSION 1 – AUTHOR RESPONSE

Reviewer: 1

Dr. Yfke P Ongena, University of Groningen

Comments to the Author:

The research described in the aims to collect qualitative and quantitative data from HCPs and people living with diabetes about their views on the future incorporation of AI into the NHS DESP. The rationale for the study is explained well and extensively. In the methods section, the development process of the questionnaire is generally clear, but it would be helpful to have a bit more details on the specific items used.

Response: We thank the reviewer for their comments. The questionnaire is currently in phase 1 of the distribution phase and is in the process of being amended based on the qualitative work detailed. Hence, we decided not to publish the survey items with this publication as the survey items are subject to change, and do not wish to bias any survey participants who would be contributing to the

survey. We aim to publish the survey items along with the results of the survey, however, the key themes of the survey which gives insights into its structure and content can be found in Table 1, page 7.

Also, the authors write that majority of questions utilised a 5-point Likert scale from “Strongly agree” to “Strongly disagree”. Such Likert-type agree-disagree (AD) scales suffer from response order effects, in the decremental AD scales (from positive agree to negative disagree), primacy effects are likely to occur (Höhne & Krebs, 2018). Especially if the items are worded positively, an acquiescence bias in responses is likely to occur. Therefore it is recommended to use an incremental scale or to use item-specific response options.

Response: The research team are aware of the potential biases involved with using Likert scales to measure responses. We agree that specific questions may also be subject to bias and there is no single perfect solution. However, Likert scales are considered by others as one of the best methods of psychometric testing ¹ and have been adopted in previous research by peers on this topic², which informed our decision to utilise this scale. Adopting Likert scales will also allow for convenient and easy-to-interpret visualisation of findings to be presented back to the public. To mitigate against the likelihood of acquiescence bias, survey items seeking to elicit perceptions of respondents have been both positively and negatively worded within the same group of questions. (additional text is now included in the last paragraph on page 7 of the manuscript). This will allow us to directly assess extent to which response acquiescence differed between positively and negatively framed questions. Importantly, all questions were codesigned with our focus groups who agreed with the questionnaire structure based on familiarity with this type of question format and they confirmed a good balance between positively and negatively worded questions. The manuscript has been amended to include the following sentence:

“Focus group participants were familiar with the use of Likert -style questions in other settings and this question style has been considered as one of the best measurement methods in psychological research (page 7).

The use of a focus group for development of a questionnaire in its first stage is obvious, but it is not clear to me why the focus group was used for final feedback on the questionnaire (“The surveys were redistributed to focus groups to ensure participants were happy with the layout and any suggested feedback was considered.”) How did this redistribution take place? What do the authors mean with “feedback was considered”?

Response: The questionnaire went through an iterative process of review and re-review with our focus groups to ensure that the questionnaire was appropriate for the target populations whilst accurately reflecting the aims and objectives of the study. This continuous feedback with our focus groups participants was seen as an important feature of the co-design process, to ensure that participants feedback had been accurately interpreted and captured. It showed participants demonstrable impact of involvement in research, improved channels of communication, facilitated focus group retention and ultimately dissemination of the survey. Redistribution took place via email and any comments received were incorporated into the next amendment of the survey until no more comments were provided. The main focus of redistribution was to affirm understanding of the survey content and to ensure that the survey would provide meaningful results for analysis and was meeting the aims and objectives of our research. The phrase “feedback was considered” has been removed from the main document and replaced with the wording below which is shown in track changes to clarify the iterative process:

“The updated surveys were redistributed to focus groups to ensure participants were happy with the layout and to ensure participant feedback had been accurately represented in the final survey version.” (page 7).

Since mobile phones were also allowed as a device for survey completion: was this also taken into account when testing usability and technical functionality of the survey platform?

Response: The usability and technical functionality of the survey platform using mobile phones was explored by the research team when deciding on the survey platform to use. Focus group participants were able to complete the survey during the design stage from their preferred devices which included mobile phones and laptops. Participants were specifically asked about the readability and layout of the survey on different devices during focus group sessions – this detail has been clarified in the manuscript in the section named ‘online survey platform’ on page 10. The appearance and functionality of the survey using different devices is also a key component of the validation stage during phase I of the survey distribution, we have modified the text on page 11 to clarify this step.

“This validation step will include discussions around comprehension of survey questions, readability, ease of use of the online platform using different devices, relevance of questions and whether additional questions are needed. The surveys will be reviewed and refined where necessary” (page 11).

On page 10, line 16 the authors write "expected to complete the survey only once"; it is unclear what this entails: was it technically possible to actually complete the survey more than once?

Response: Due to limitations with the survey platform used and the recruitment strategy, it would technically be possible for participants to complete the survey more than once as the survey is anonymous so the identity of potential participants is not recorded within the survey and there is no individualisation for each participant in terms of the survey link. We felt anonymity was a priority in this survey. We are aware that this is a potential limitation to the research, hence no incentives to participate have been offered so that there are no contributing factors to encourage participants to respond more than once. We believe the likelihood of anyone completing the survey more than once is highly unlikely.

Recruitment of participants in phase I is not entirely clear to me. Posters will be used, but how to make sure participants will have attention for these posters? This might introduce selection effects, depending on one's attentiveness to posters and experience in using QR-codes (or reluctance to do so). In the flow chart of figure 1 it is also mentioned that surveys will be distributed via text or email, but this is not further described in the main text.

Response: Further information on the survey recruitment strategy based on Figure 1A has been provided in the main text on page 11 in tracked changes to detail how NHS staff will be recruited via email and how people living with diabetes will be recruited by the text messaging service. Please see below for added text:

“As shown in Figure 1, NHS staff and people living with diabetes from the NEL DESP will be recruited by the clinical lead or programme manager. For NHS staff, they will be recruited by an email invitation to complete the survey, whilst people living with diabetes will be recruited by a text messaging service sent by the programme manager which is an established form of communication with patients as it is used to send appointment reminders and service feedback surveys.”

The poster contains a QR code and URL for the survey to help to prompt people living with diabetes to complete the survey, but this is a supplementary recruitment strategy as the main recruitment method is via direct contact from the screening programme manager.

In phase IV, it is not clear to me what is meant by "effectiveness of the survey".

Response: Phase IV details the use of evaluative qualitative interviews to determine whether the survey meets its aims and objectives and thus establish its effectiveness. We have amended the manuscript to clarify the purpose of phase IV, please see below:

"It is important to evaluate the survey among participants that have completed the survey so that if the research were to be repeated elsewhere or on a larger scale, any additional feedback could be considered and incorporated in future repeat surveys." (page 12)

Rather than extending the data collection period to 4 months (page 11, line 49/50) it could also be considered to increase response rates via mixed-mode surveys, for instance, paper-and-pencil versions for respondents reluctant to use electronic surveys. In addition, use of incentives (Pieper et al., 2018) or personalisation (VanGeest & Johnson, 2012) may be considered.

Response: We acknowledge and agree with the reviewer's comments and understand that a mixed-mode survey would improve response rates. We are currently considering other methods for distribution to optimise completion as part of on-going research, and have now highlighted this in the strengths and limitations section of the discussion, page 14. Please see below for the added text.

Limitations

Due to the methods used to recruit people living with diabetes and NHS staff, we are aware that there will be limitations in terms of potential selection bias and representativeness of the survey population. This includes potential underrepresentation of people where English is not their first language, unless support can be offered by family members and friends to assist with survey completion. There may also be underrepresentation from people who do not have access to the internet or an electronic device to complete the survey, which may have a greater impact on older adults. Utilising electronic distribution and completion of surveys may impact on survey response rates compared with using electronic and paper survey formats for distribution and completion.

We thank the reviewer for citing references by Pieper et al. (2018) and VanGeest and Johnson (2012) who reported financial incentives do not impact response rates to surveys in the clinical setting. We also decided not to offer financial incentives or a prize draw for survey completion as this could increase the likelihood of multiple responses from individual participants. In addition, as we are aiming to recruit 300 people living with diabetes for our patient survey as well as 300 healthcare professionals to our staff survey, it would not be financially possible to offer individual vouchers to each respondent due to budgetary limitations.

Personalisation has been incorporated into the specified recruitment strategies to encourage participation, by utilising communication methods that are already pre-established and familiar to patients, for instance using text messaging services sent by the screening programme manager or using handouts which will be distributed by NHS staff after diabetic eye screening appointments. This detail has been added to the manuscript on page 11, please see below for the added text:

"Recruitment of participants will take place either via text messaging service sent by the programme manager, or by a handout with the survey URL and QR code which will be distributed by NHS staff during appointments."

References:

Höhne, J. K., & Krebs, D. (2018). Scale direction effects in agree/disagree and item-specific questions: A comparison of question formats. *International Journal of Social Research Methodology*, 21(1), 91-103.

Pieper, D., Kotte, N., & Ober, P. (2018). The effect of a voucher incentive on a survey response rate in the clinical setting: a quasi-randomized controlled trial. *BMC medical research methodology*, 18(1), 1-4.

VanGeest, J. B., & Johnson, T. P. (2012). Using incentives in surveys of cancer patients: do “best practices” apply?. *Cancer Causes & Control*, 23, 2047-2052.

Reviewer: 2

Dr. Ehsan Vaghefi, The University of Auckland, The University of Auckland

Comments to the Author:

This is a clear description of the survey that is going to be performed in the community.

Response: We thank reviewer 2 for their time taken to review our paper and for their positive feedback.

1. Jebb AT, Ng V, Tay L. A Review of Key Likert Scale Development Advances: 1995–2019. *Frontiers in Psychology* 2021;12
2. Yap A, Wilkinson B, Chen E, et al. Patients Perceptions of Artificial Intelligence in Diabetic Eye Screening. *Asia Pac J Ophthalmol (Phila)* 2022;11(3):287-93. doi: 10.1097/APO.0000000000000525 [published Online First: 2022/07/01]